Efficacy of 4% articaine vs 2% lidocaine in mandibular and maxillary block and infiltration anaesthesia in patients with irreversible pulpitis: a systematic review and meta-analysis

Miglani Sanjay 1
Ansari Irfan 1
Patro Swadheena 2
Mohanty Ankita 2
Mansoori Shahnaz 3
Ahuja Bhoomika 4
Karobari Mohmed Isaqali 5 6
Shetty Krishna Prasad 7 8
Saeed Musab Hamed 7 8
Luke Alexander Maniangat a.luke@ajman.ac.ae 7 8
Pawar Ajinkya M. ajinkya@drpawars.com 9
1 Department of Conservative Dentistry & Endodontics, Faculty of Dentistry, Jamia Millia Islamia (A Central University), Okhla, New Delhi, India , Delhi , India
2 Department of Conservative Dentistry & Endodontics, Kalinga Institute of Dental Sciences, Bhubaneswar, Odisha, India , Bhubaneswar , India
3 Department of Public Health, Faculty of Dentistry, Jamia Millia Islamia (A Central University), Okhla, New Delhi, Delhi, India , Delhi , India
4 Department of Paediatric Dentistry, K D Dental College, Mathura, Uttar Pardesh, India , Mathura , India
5 Conservative Dentistry Unit, School of Dental Sciences, Universiti Sains Malaysia, Health Campus, Kubang Kerian, Kota Bharu, Kelantan, Malaysia , Kota Bharu , Malaysia
6 Department of Conservative Dentistry & Endodontics, Saveetha Dental College & Hospitals, Saveetha Institute of Medical and Technical Sciences University , Chennai , Tamil Nadu , India
7 Department of Clinical Science, College of Dentistry, Ajman University , Al-Jurf Ajman , UAE
8 Centre of Medical and Bio-allied Health Sciences Research, Ajman University , Ajman , UAE
9 Department of Conservative Dentistry & Endodontics, Nair Hospital Dental College, Mumbai, Maharashtra, India , Mumbai , India
Testarelli Luca
Electronic publication date: 2021 Sep 24
Publication date: 2021
Volume: 9
Electronic Location ID: e12214
Received 2021 May 12; Accepted 2021 Sep 5
Copyright: ©2021 Miglani et al.
Copyright year: 2021
Copyright holder: Miglani et al.
License: This is an open access article distributed under the terms of the Creative Commons Attribution License, which permits unrestricted use, distribution, reproduction and adaptation in any medium and for any purpose provided that it is properly attributed. For attribution, the original author(s), title, publication source (PeerJ) and either DOI or URL of the article must be cited.
License URL: https://creativecommons.org/licenses/by/4.0/

Keywords: Articaine, Irreversible pulpitis, Lidocaine, Meta-analysis, Sensitivity analysis, Success rate

Funding: The authors received no funding for this work.

==============================
Objective

The goal of this systematic review and meta-analysis is to determine the performance of 4% Articaine vs. 2% Lidocaine for mandibular and maxillary block and infiltration anaesthesia in patients with irreversible pulpitis (IP).

Methods

PubMed/MEDLINE, Cochrane Central Register of Controlled Trials, Web of Science, Google Scholar, and Open Gray were used to conduct a thorough literature search. A manual search of the reference lists of the publications found was also carried out. Two reviewers critically evaluated the papers for inclusion and exclusion criteria, and data extraction was done on the selected publications. The Cochrane Collaboration Tool and the Minors checklist were used to assess the quality of the selected studies for randomised controlled trials (RCTs) and non-randomised studies, respectively. The RevMan software was used to perform a meta-analysis of the pooled data and subgroups according to the technique of anaesthetic solution delivery, as well as a sensitivity analysis (P < 0.05).

Results

A total of twenty-six papers were included in the qualitative synthesis, with twenty-two of them being included in the meta-analysis. There were fifteen studies with a low potential for bias, three with a moderate potential for bias, and seven with a high potential for bias. The combined results of the 19 trials in the tooth level unit revealed that 4% articaine had a success rate 1.37 times greater than 2% lidocaine for mandibular teeth (RR, 1.37; 95% CI [1.17–1.62]; P = 0.0002). For the maxillary buccal infiltration method, the combined results from the three trials revealed that 4% articaine resulted in a success rate 1.06 times greater than 2% lidocaine (RR, 1.06; 95% CI [0.95–1.2]; P = 0.3). Excluding subgroups with a single study in sensitivity analysis for mandibular teeth revealed a substantial improvement in the success rate of the articaine group in treating IP when compared to the lidocaine group.

Conclusion

The findings of this meta-analysis back up the claim that articaine is more effective than lidocaine in providing anaesthesia in patients with IP. PROSPERO Registration No.: CRD42020204606 (https://www.crd.york.ac.uk/prospero/display_record.php?ID=CRD42020204606).

Introduction

Dental caries is the most widespread non-communicable disease and a major public health concern globally. It is also the most common preventable disease, recognised as the primary cause of oral pain and tooth loss (World Health Organization, 2017). Pulpitis is the inflammation of the dental pulp and a sequel to caries. It is clinically classified as reversible or irreversible (Rôças et al., 2015). Irreversible pulpitis (IP), usually develops when the pulp is exposed to caries biofilm causing necrosis or death of pulp tissues, indicating the need for endodontic treatment (Li et al., 2012; Zanjir et al., 2019).

Successful anaesthesia is a hallmark of painless endodontic treatment. It not only keeps the patient calm and relaxed but also allows the dentist to perform the endodontic procedure with ease (Subbiya & Pradeepkumar, 2016; Howait & Basunbul, 2019). Pulpal anaesthesia commences with the administration of traditional nerve blocks (NBs) or infiltration anaesthesia. For maxillary teeth, NB injections are administered at the anterior superior alveolar (ASA), middle superior alveolar (MSA), and/or posterior superior alveolar (PSA) nerves. For mandibular anterior teeth and molars, both the inferior alveolar nerve block (IANB) and buccal infiltration (BI) injections are administered (Reader, Nusstein & Hargreaves, 2011). Local anaesthetic (LA) drugs have a peripheral effect and block the transmission of nerve impulses. Factors that affect anaesthetic drug efficacy are the type of applied drug, correct injection site, injection velocity, and amount and dosage of the injected drug. Also, the presence of inflammation at the injection site should not be overlooked (Modaresi et al., 2016).

The success rate of achieving deep pulpal anaesthesia lowers in patients with IP. The success rate of IANB can be reduced to <30% (Shahi et al., 2018), and that of maxillary NBs to <60% (Sherman et al., 2008). It is broadly accepted that achieving anaesthesia in patients with IP is more complex, as compared to normal, healthy pulps (Dou et al., 2013; Dou et al., 2018). Inflamed pulp shows lower pH levels, lowering the penetration of basic anaesthetic into the nerve membrane, thus delaying or preventing pulpal anaesthesia (Zanjir et al., 2019). This state of the tooth is frequently referred to as a ‘hot pulp’, which requires supplementary approaches to ensure a pain-free treatment (Subbiya & Pradeepkumar, 2016). Nonetheless, the effect of different anaesthetic agents and techniques along with or without supplemental infiltration needs to be assessed.

Lidocaine, the first commercialised amide LA solution shows rapid onset when used for most of the dental treatments and is considered as the gold standard LA agent due to its high efficacy, low allergenicity and minimal toxicity (Su et al., 2016). Articaine [(4-methyl-3-[1-oxo-2-(propylamino)-propionamido]-2-thiophene-carboxylic acid methyl ester hydrochloride)] is a unique amide LA agent that contains a thiophene ring instead of a benzene ring, demonstrating increased liposolubility and high tissue penetrability as compared to lignocaine. The thiophene ring raises the diffusion of the anaesthetic solution into the cortical bone, thereby penetrating the mandibular dense cortical bone as well as maxillary cortical plates. Previous studies by De Geus et al., (2020) and Srinivasan et al. (2017) have also shown that articaine is equally effective in comparison to other anaesthetics with the success rate ranging from 64 to 87%.

Systematic reviews comparing the anaesthetic efficacy of articaine and lidocaine for dental procedures have been published. Kung, McDonagh & Sedgley (2015) concluded that articaine has a significant advantage over lidocaine as a supplementary infiltration after mandibular block anaesthesia but no advantage when used alone as mandibular block or maxillary infiltration anaesthesia. Su et al. (2016) stated that at the injection phase and treatment phase, 4% articaine is superior in controlling pain and increasing the success rate of local anaesthesia than 2% lidocaine. However, out of 24 articles included in that analysis, 20 articles were Chinese language reports which could not be accessed and retrieved from the databases by the authors of this review. Both the reviews described above were based on searches conducted until 2013. De Geus et al. (2020) in the network meta-analysis concluded that in patients with IP, the use of articaine increased the IANB success rate. This review considered the efficacy of IANB in patients with IP and did not include maxillary injection techniques as well as other supplemental techniques. A preliminary electronic database search revealed that since the publication of the above reviews, several new randomised clinical trials (RCTs) comparing articaine and lidocaine for patients with symptomatic IP have been published. Therefore, the present systematic review assessed the efficacy of 4% articaine vs 2% lidocaine in the mandibular and maxillary block and infiltration anaesthesia in patients with IP.

Methods

Protocol development

This systematic review and meta-analysis are written and reported according to the Preferred Reporting Items for Systematic Review and Meta-Analyses (PRISMA) statement and registered in PROSPERO (CRD42020204606). The following focused question in the Patient, Intervention, Comparison, and Outcome (PICO) format was proposed “Is there a difference in the efficacy of 4% articaine vs 2% lidocaine in the mandibular and maxillary block and infiltration anaesthesia in patients with IP?”

Search strategy

To obtain publications in the English language, a complete electronic search was conducted through July 2020 on databases such as PubMed and MEDLINE, Cochrane Central Register of Controlled Trials, and Web of Science. A detailed electronic search of the journals listed in Table 1 was carried out. The searches in the clinical trials database, cross-referencing and Grey literature were conducted using Google Scholar, Greylist, and OpenGrey.

Table 1 The search strategy and PICOS tool.

Search strategy		
Focused Question	Is there a difference in the efficacy of 4% Articaine versus 2% Lidocaine in mandibular and maxillary block and infiltration anaesthesia in patients with irreversible pulpitis?	
Search strategy		
Population (#1)	((Irreversible pulpitis [Text Word]) OR (”maxilla”[MeSH Terms] OR maxillary[Text Word]) AND (”tooth”[MeSH Terms] OR teeth[Text Word]) OR (”mandible”[MeSH Terms] OR mandibular[Text Word]) AND (”tooth”[MeSH Terms] OR teeth[Text Word]) OR lower teeth [Text Word] OR upper teeth [Text Word] OR ”molar”[MeSH Terms] OR molar[Text Word] OR posterior teeth [Text Word] OR anterior teeth [Text Word] OR premolar [Text Word] OR ”incisor”[MeSH Terms] OR incisor[Text Word] OR canine [MeSH]))	
Intervention (#2)	((Articain [Text Word] OR Articaine [Text Word] OR Carticaine [MeSH] OR Carticaine Hydrochloride [Text Word] AND (Local Anesthesia [Text Word] OR Infiltration Anesthesia [Text Word] OR nerve block [Text Word] OR inferior alveolar nerve block [Text Word] OR buccal infiltration [Text Word] OR Infra-orbital nerve block [Text Word] OR Anterior superior nerve block [Text Word] OR Middle superior nerve block [Text Word]))	
Comparisons (#3)	((Lidocaine [MeSH] OR Lidocaine Hydrochloride [Text Word] OR Lignocaine [Text Word]) AND (Local Anesthesia [Text Word] OR Infiltration Anesthesia [Text Word] OR nerve block [Text Word] OR inferior alveolar nerve block [Text Word] OR buccal infiltration [Text Word] OR Infra-orbital nerve block [Text Word] OR Anterior superior nerve block [Text Word] OR Middle superior nerve block [Text Word]))	
Outcomes (#4)	(Success [Text Word] Pain [Text Word] OR onset time [Text Word] OR duration [Text Word] OR Visual analogue scale [MeSH] OR Heft Parker Visual Analog Scale [Text Word])	
Study design (#5)	(Clinical trials [MeSH] OR randomized controlled studies [Text Word] OR randomized control trials [MeSH] OR randomized control clinical trial MeSH OR non-randomized control trials [Text Word] OR Quasi experimental studies [Text Word] OR before and after study design [Text Word] OR cohort studies [Text Word] OR in vivo study [Text Word])	
Search combination	#1 AND #2 AND #3 AND #4 AND #5	
Database search		
Language	No restriction (Articles in English language or other language where English translation is possible.)	
Electronic databases	PubMed/MEDLINE, Cochrane Central Register of Controlled Trials, Web of Science	
Journals	Journal of Endodontics, International Endodontic Journal, Australian Endodontic Journal, Clinical Oral Investigations, Journal of Conservative Dentistry, Journal of American Dental Association	
Period of publication	Studies published between 1-1-2011 to 30-09-2020	

Articles were found using Medical Subject Headings (MeSH) terms, keywords, and other free terms coupled with Boolean operators (OR, AND). Following the syntactic guidelines of each database, the same terms were utilised across all search platforms. Table 1 shows the search strategy, population, interventions, comparisons, outcomes, and study design (PICOS) tool.

Inclusion criteria outline according to the PICOS strategy

Population (P):

Studies on patients with symptomatic IP, requiring endodontic treatment under maxillary and mandibular infiltration or blocked anaesthesia on at least one tooth in the mandibular or maxillary region irrespective of age, gender, race, or socioeconomic status, were evaluated. Active responses to spontaneous pain, thermal tester, cold tester, or an electronic pulp tester were considered as diagnostic criteria for IP.

Interventions (I): Studies using 4% articaine as a LA solution for the treatment of IP.

Comparison (C): Studies using 2% lidocaine as a LA solution for the treatment of IP.

Outcome (O): Primary outcome: It included the success rate assessed as ‘no/mild’ pain during access cavity preparation and biomechanical preparation phase on the Heft Parker visual analogue scale (VAS).

Secondary outcome: Onset of anaesthesia assessed from the time lapse between the end of the NB and the onset of symptoms of subjective anaesthesia (feeling of heaviness at the site of injection) were calculated in seconds, and the pain was assessed quantitatively during the treatment.

Study design (S): In vivo studies, clinical trials, RCTs, cluster randomised trials, quasi-experimental studies and non-randomised trials (NRS).

Exclusion criteria

• Studies involving patients with a significant medical history or on medicaments that may affect the anaesthetic assessment.

• Observational study designs, case reports, case series, cross-sectional studies, and reviews.

• Trials reporting a single intervention.

• Article reporting only abstracts and full-texts were not available in the database.

Screening process

Two reviewing authors did the search and screening according to the previously stated methodology (SP & AM). The titles and abstracts were examined first. Second, full-text publications were picked for in-depth reading and analysis based on the data extraction criterion’s inclusion and exclusion criteria. Cohen’s kappa (k) determined the degree of agreement between the two reviewers to be 0.94 for titles and abstracts and 0.96 for full-texts. After discussions, the third author SM was able to reconcile the disagreements among the authors/reviewers. The authors of the listed papers were contacted via email for clarification of any concerns or missing data.

Data extraction

The two independent authors (IA & BA) extracted the following data from the included studies: author names, study design, tooth, sample size, method of pulp testing, type of local anaesthesia used, injection technique, method of analysis, method of outcome assessment, follow-up and author’s conclusions.

Assessments of the risk of bias and quality

The level of evidence for included studies was assessed using the Joanna Briggs Institute (JBI) level of evidence (The Joanna Briggs Institute, 2014). The quality of the selected studies was assessed using the Cochrane Collaboration Tool (Higgins et al., 2019) for RCTs, including random sequence generation, allocation concealment, blinding of participants, incomplete outcome data, selective reporting, and other biases. The quality of NRS was assessed using the Minors checklist (Slim et al., 2003), wherein the minimum outcome assessment time of 5 min was considered appropriate for the included studies.

Statistical analysis

Review Manager (RevMan) 5.3 was used for statistical analysis. The combined results were expressed as relative risks (RRs) for the dichotomous data at 95% confidence intervals (CIs) and P < 0.05 was considered significant. Statistical heterogeneity was assessed by the I2 test at α = 0.10. Subgroup analysis was conducted for I2 > 50% and P ≤ 0.10. For I2 > 50%, the random-effects model was applied. Sensitivity analysis was conducted to assess the stability of the results. Funnel plots were drawn to detect publication bias for studies exceeding 10 in number for each outcome assessed (Su et al., 2016).

Results

Literature search

The PRISMA statement flowchart summarising the selection process is presented in Fig. 1. Among 33 full-text articles, 26 were selected after pre-screening, applying the eligibility criteria, and addressing the PICOS question. Seven studies were excluded as three did not have a control group, three had inappropriate population variables, and one applied intraosseous injection technique, hence only 26 studies were included in the qualitative analysis, whereas 22 out of 26 studies for quantitative synthesis.

Figure 1 PRISMA flow diagram.

Study characteristics

The general characteristics of 26 studies (Gao & Meng, 2020; Kumar et al., 2020; Aggarwal et al., 2019; Ghazalgoo et al., 2018; Martínez-Martínez, Freyle-Granados & Senior-Carmona, 2018; Shapiro et al., 2018; Aggarwal, Singla & Miglani, 2017; Umesh, 2017; Allegretti et al., 2016; Hosseini et al., 2016; Saraf et al., 2016; Zain et al., 2016; Monteiro et al., 2015; Rajput et al., 2015; Ahmad et al., 2014; Nabeel, Ahmed & Sikander, 2014; Rogers et al., 2014; Sood, Hans & Shetty, 2014; Ashraf et al., 2013; Tortamano et al., 2013; Kanaa, Whitworth & Meechan, 2012a; Kanaa, Whitworth & Meechan, 2012b; Aggarwal et al., 2011; Poorni et al., 2011) are presented in Table 2. All included studies were unicentric trials published between 2011 and 2020. Of the 26 studies, 10 studies (Kumar et al., 2020; Aggarwal et al., 2019; Lokhande et al., 2019; Aggarwal, Singla & Miglani, 2017; Umesh, 2017; Saraf et al., 2016; Sood, Hans & Shetty, 2014; Subbiya et al., 2011; Aggarwal et al., 2011; Poorni et al., 2011) were conducted in different parts of India, 3 in Iran (Ghazalgoo et al., 2018; Hosseini et al., 2016; Ashraf et al., 2013), Pakistan (Zain et al., 2016; Rajput et al., 2015; Nabeel, Ahmed & Sikander, 2014) and Brazil (Allegretti et al., 2016; Monteiro et al., 2015; Tortamano et al., 2013), 2 in the USA (Shapiro et al., 2018; Rogers et al., 2014), 2 in the UK (Kanaa, Whitworth & Meechan, 2012a; Kanaa, Whitworth & Meechan, 2012b) and one each in China (Gao & Meng, 2020), Colombia (Martínez-Martínez, Freyle-Granados & Senior-Carmona, 2018) and Saudi Arabia (Ahmad et al., 2014). The study design of 24 studies (Gao & Meng, 2020; Kumar et al., 2020; Aggarwal et al., 2019; Ghazalgoo et al., 2018; Martínez-Martínez, Freyle-Granados & Senior-Carmona, 2018; Shapiro et al., 2018; Aggarwal, Singla & Miglani, 2017; Umesh, 2017; Allegretti et al., 2016; Hosseini et al., 2016; Saraf et al., 2016; Zain et al., 2016; Monteiro et al., 2015; Rajput et al., 2015; Nabeel, Ahmed & Sikander, 2014; Rogers et al., 2014; Sood, Hans & Shetty, 2014; Ashraf et al., 2013; Kanaa, Whitworth & Meechan, 2012a; Kanaa, Whitworth & Meechan, 2012b; Aggarwal et al., 2011; Poorni et al., 2011) was RCTs, and the remaining 2 studies (Ahmad et al., 2014; Subbiya et al., 2011) were NRS. The age of the participants was 15–65 years. The ethical clearance and informed consent were obtained in all except 2 studies (Lokhande et al., 2019; Subbiya et al., 2011). A total of 1,824 teeth with completely formed root apex, diagnosed with symptomatic IP and anaesthetised with either articaine or lidocaine were included in the review, 1,578 mandibular teeth from 22 studies (Gao & Meng, 2020; Kumar et al., 2020; Aggarwal et al., 2019; Ghazalgoo et al., 2018; Martínez-Martínez, Freyle-Granados & Senior-Carmona, 2018; Shapiro et al., 2018; Aggarwal, Singla & Miglani, 2017; Umesh, 2017; Allegretti et al., 2016; Zain et al., 2016; Monteiro et al., 2015; Rajput et al., 2015; Ahmad et al., 2014; Rogers et al., 2014; Sood, Hans & Shetty, 2014; Ashraf et al., 2013; Tortamano et al., 2013; Kanaa, Whitworth & Meechan, 2012b; Subbiya et al., 2011; Aggarwal et al., 2011; Poorni et al., 2011) and 246 maxillary teeth from 4 studies comprised the present meta-analysis (Hosseini et al., 2016; Saraf et al., 2016; Nabeel, Ahmed & Sikander, 2014; Kanaa, Whitworth & Meechan, 2012a) (Table 2).

Table 2 Characteristics of the included studies.

Study Id	Place of study	Study Design	Sampling Technique	Sample Size Intervention /Control	Age group in years Intervention /Control Mean(SD) or Range	Gender M/F Intervention /Control N	Teeth assessed	Pulp testing	Periapical radiolucency/ widened periodontal ligament	Anaesthetic solution	Anaesthesia technique	Solution dosage in	Recording time in minutes	Outcome assessed	Authors conclusions	
										Intervention group	Control group	Intervention group	Control group	Intervention group	Control group				
Gao & Meng (2020)	China	Prospective, randomised clinical study	Random numbers table	52/52	39.2(13.2)/ 40.1(12.9)	24 M, 28 F/ 22 M, 30 F	Mandibular posterior teeth	Electric pulp test Cold test.	No/-	4% Articaine with 1:100,000 adrenaline	2% Lidocaine with 1:100,000 adrenaline	Supplementary intraligamentary BI after IANB failure	Supplementary intraligamentary BI after IANB failure	0.9	0.9	5	Success of anaesthesia	Supplemental BI with Articaine following IANB can be considered a more successful anaesthetic agent in mandibular posterior teeth with irreversible pulpitis compared with lidocaine	
Kumar et al. (2020)	India	Double blind randomized clinical study	Randoy assigned	13/ 12	15–55	–	Symptomatic mandibular posterior teeth	Thermal tests with endo frost and heated gutta-percha sticks Electric pulp test	No/Yes	4% Articaine with 1:100,000 Epinephrine	2% Lidocaine; 1:80,000 Epinephrine	IANB	IANB BI	1.8	1.8	15	IANB success BI Success	Overall success rate with 4% Articaine was 92% and with 2% Lidocaine was 75% after IANB and BI.	
Aggarwal et al. (2019)	India	Randomized, double-blind clinical trial	Online random generator using a permuted block stratified randomization	41/41	37(8) / 34(9)	24 M 17 F/ 27 M 14 F	Symptomatic mandibular molars	Cold tests Electric pulp tests	No/-	4% Articaine with 1:100,000 epinephrine	2% Lidocaine with 1:80,000 epinephrine	Supplementary intraligamentary injections after IANB failure	Supplementary intraligamentary injections after IANB failure	0.6	0.6	5	Success of anaesthesia	2% lidocaine with 1:80,000 epinephrine and 4% Articaine with 1:100,000 epinephrine as supplementary intraligamentary injections after an unsuccessful primary IANB improved the success rates, with no significant difference between them.	
Ghazalgoo et al. (2018)	Iran	Prospective double-blind clinical trial study	Random	44/44	–	–	Symptomatic mandibular first molar	Cold test	–	4% Articaine with 1:100,000 epinephrine	2% Lidocainewith 1:100,000 epinephrine	IANB	IANB	–	–	15	Pain levels at 0, 2, 4, 6, 12, 18, 36, and 48 h	Articaine for IANB may increase post-root canal treatment comfort than lidocaine	
Lokhande et al. (2019)	India	Single blinded randomized clinical trial	Convenience	30/30	Above 18 years	–	Symptomatic mandibular molar	–	No/-	4% Articaine (1:100000 adrenaline)	2% Lidocaine (l:80000 adrenaline)	BI combined with intraligamentary injection	BI combined with intraligamentary injection	1.8 + 0.2	1.8 + 0.2	5	Anaesthesia success	BI with 4% Articaine along with supplemental injection (intraligamentary) increased anaesthetic success rates.	
Martínez-Martínez, Freyle-Granados & Senior-Carmona (2018)	Colombia	Randomized, double-blind, parallel-controlled clinical trial	Block randomization	18/18	Over 18	–	Lower molars	Vitalometer	–	4% Articaine with 1:100,000 epinephrine	2% Lidocaine with 1:80,000 epinephrine	IANB	IANB	1.8	1.8	10	Anaesthesia success	No statistically significant differences were found in the anaesthetic efficacy of 2% lidocaine and 4% Articaine in lower molars with vital pulp. Articaine showed a better anaesthetic success rate.	
Shapiro et al. (2018)	United States	Prospective, randomized, double-blind	Block randomization	76/73	37–41	41 M, 35 F/ 32 M, 41 F	Symptomatic mandibular molar	Cold testing with Endo-Ice	No/Yes	4% Articaine with 1:100,000 epinephrine	2% Lidocaine with 1:100,000 epinephrine	Supplementary BI after IANB failure	Supplementary BI after IANB failure	1.7	1.7	5	Successful infiltration anaesthesia	Supplemental BI with 4% Articaine and 2% lidocaine was found to have comparable success in the first molar region. BI with 4% Articaine was significantly more effective than 2% lidocaine for mandibular second molars with irreversible pulpitis.	
Aggarwal, Singla & Miglani (2017)	India	Prospective, double-blind clinical study	Randomized using an online random generator	32/31	34(6.5)/ 37(8.3)	16 M, 14F/ 22 M, 9F	Mandibular molar	Pulp sensitivity tests	No/-	4% Articaine with 1:100,000 epinephrine	2% Lidocaine with 1:200,000 epinephrine	IANB	IANB	Standard 4% Articaine cartridge/ 1.8 mL	1.8	15	Success of anaesthesia	The 2% lidocaine solution used for IANB had similar success rates when compared with 4% Articaine	
Umesh (0000)	India	Prospective, randomized, triple-blind study	Randomly- Sequence generated by computerized permutted block	30/30	18-65	–	Symptomatic mandibular molars	Electric pulp test Thermal test	–	4 % Articaine with 1:1 , 00,000 Adrenaline	2% Lignocaine with 1: 80,000 Adrenaline	IANB	IANB	6	3	–	Pre-post-operative pain	4% articaine + 1:1,00,000 epinephrine performed better than 2% lignocaine + 1:80,000 epinephrine in reducing pain during endodontic access opening and instrumentation.	
Allegretti et al. (2016)	Brazil	Prospective, randomized, double-blind clinical study	Simple random	22/22	28.7/ 30.3	10 M, 12 F/ 9 M, 13 F	Symptomatic first or second molars	Electric pulp test Cold testing with Endo-Frost	No/-	4% Articaine with 1:100,000 epinephrine	2% Lidocaine with 1:100,000 epinephrine	Standard IANB	Standard IANB	3.6	3.6	14 to 16	Success of anaesthesia	Neither of the solutions resulted in 100% anaesthetic success in patients with irreversible pulpitis of mandibular molars.	
Hosseini et al. (2016)	Iran	Prospective, randomized double-blind study	Simple random	25/25	–	–	Asymptomatic first maxillary molar	Eelectric Pulp Tester Cold tests	No/-	4% Articaine with 1:100000 epinephrine	2% Lidocaine with 1:80000 epinephrine	BI	BI	1.8	1.8	5	Success of anaesthesia	The type of anaesthetic solution had no significant influence on the success rate of anaesthesia with Articaine and lidocaine being similarly effective.	
Saraf et al. (2016)	India	Clinical study	Random	20/20	–	–	Symptomatic maxillary anteriors and premolars	Electric pulp tester	–	Group I: Articaine HCl 4% with 1:100,000 adrenaline Group II: Articaine HCl 4% with 1:100,000 adrenaline	Group III: Lidocaine HCl 2% with 1:80,000 adrenaline Group IV: Lidocaine HCl 2% with 1:80,000 adrenaline	Group I: Anterior middle superior alveolar nerve block Group II: Infraorbital nerve block	Group III: Anterior middle superior alveolar nerve block Group IV: Infraorbital nerve block	Group I: 0.6–1.4 Group II: 0.9–1.2	Group III: 0.6–1.4 Group IV: 0.9–1.2	30	Onset of anaesthesia Pain assessment	Articaine 4% proved to be more efficacious than lidocaine 2%, and AMSANB was more advantageous than IONB in securing anaesthesia of maxillary anteriors and premolars	
Zain et al. (2016)	Pakistan	Prospective and randomized clinical trial	Lottery method	78/78	31.46(10.99)	46 M, 32 F/ 46M, 32 F	Symptomatic mandibular 1st molar	yes	No/-	4% Articaine with 1:100,000 epinephrine	2% lidocaine with 1:100,000	BI	IANB	1.8	1.8	10	Success of anaesthesia	4% Articaine BI can be considered a viable alternative to 2% lidocaine IANB in securing successful pulpal anaesthesia for endodontic therapy.	
Monteiro et al. (2015)	Brazil	Prospective and randomized clinical trial	Random numbers	30/20	28(13.8)/ 33.5(16.5)	5 M, 25 F/ 4 M, 16 F	Symptomatic mandibular molars	Cold tests	No/ Yes	4% Articaine with 1: 100 000 epinephrine	2% Lidocaine with 1: 100 000 epinephrine	BIs	IANB injections	1.8	1.8	10	Success of anaesthesia	Single anaesthesia techniques (IANB or BI) were not able to achieve pain-free emergency endodontic treatment.	
Rajput et al. (2015)	Pakistan	Prospective, randomized clinical trial	Simple randomized	30/30	18–65	–	Symptomatic mandibular first molar	–	–	4% Articaine with 1:100,000 epinephrine	2% Lidocaine with 1:200,000 epinephrine	Standard BI	Standard IANB	1.7	1.8	10	Success of anaesthesia	4% Articaine with 1:100,000 epinephrine can be considered as an alternative for pulpal anaesthesia in mandibular first molar with irreversible pulpitis.	
Ahmad et al. (2014)	Riyadh, Saudi Arabia	Non-randomized control trial	–	15/15	18–40	–	Symptomatic mandibular teeth	Cold testing with an ice stick Electric pulp tester	No/-	4% Articaine with 1:100000 epinephrine	2% Lidocaine with 1:200000 epinephrine	Standard IANB	Standard IANB	–	–	15	Success of anaesthesia	4% Articaine with 1:100,000 epinephrine showed better anaesthetic effect when administered as inferior alveolar nerve block as compared to 2% lidocaine with 1:200,000 epinephrine.	
Nabeel, Ahmed & Sikander (2014)	Pakistan	Randomized clinical trial	Computer- generated list of random numbers	38/38		15 M, 23 F/ 18 M, 20 F	Maxillary first premolars	–	–	4% Atricaine with 1:100,000 epinephrine	2% Lidocaine with 1:100,000 epinephrine	BI	BI	1.7	1.7	5	Successful infiltration anaesthesia	The anaesthetic efficacy of Articaine is comparable to that of Lidocaine in subjects with acute irreversible pulpitis of maxillary teeth with irreversible pulpitis.	
Rogers et al. (2014)	United States	Prospective, double-blind, randomized, controlled clinical trial	Block randomization	39/35	36(14)/ 36 (12)	17 M, 22 F / 12 M, 23 F	Symptomatic mandibular molar	Cold testing with Endo-Ice	No/Yes	4% Articaine with 1:100,000 epinephrine	2% Lidocaine with 1:100,000 epinephrine	Supplementary BI after IANB failure	Supplementary BI after IANB failure	1.7	1.7-	5	Successful infiltration anaesthesia	Supplemental BI with Articaine was significantly more effective than Lidocaine.	
Sood, Hans & Shetty (2014)	India	Prospective, randomized, double-blind clinical study	Not mentioned	50/50	26.46/28.90	20 M/30 F, 27 M/23 F	Mandibular posterior teeth	Electric pulp testing Cold testing using Roeko Endo-Frost	No/Yes	4% Articaine with 1:100,000 epinephrine	2% Lidocaine with 1:80,000 epinephrine	IANB	IANB	1.8	1.8	10	Absence/presence of pain	No difference in the efficacy of both the dental anaesthetic agents in controlling pain during the treatment of irreversible pulpitis.	
Ashraf et al. (2013)	Iran	Prospective, randomized, double-blind study	Randomized using random allocation software	58/58	37.9 (10.0)/ 32.5 (8.7)	24 M,27 F/ 23 M, 28 F	Symptomatic first or second mandibular molar	Cold testing by using an ice stick	No/Yes	4% Articaine with 1:100,000 epinephrine	2% Lidocaine with 1:100,000 epinephrine	Standard IANB and long buccal injections	Standard IANB and long buccal injections	1.5 + 0.3	1.5 + 0.3	5	Successful infiltration anaesthesia	Articaine seems to raise anaesthetic success more effectively compared with lidocaine after an incomplete IANB is supplemented with an infiltration injection by using the same anaesthetic for both injections in teeth with irreversible pulpitis.	
Tortamano et al. (2013)	Brazil	Prospective, randomized, double-blinded clinical study	Random	20 (10 each)/ 10	–	–	Asymptomatic mandibular posterior molars	Electric pulp stimulator	No/-	ARTI100–4% Articaine with 1:100,000 epinephrine	2% lidocaine with 1:100,000 epinephrine	IANB	IANB	1.8	1.8	10	Onset of pulpal anaesthesia	4% Articaine with 1:100,000 epinephrine exhibited faster onset and also had longest duration of pulpal anaesthesia when compared with all solutions	
Kanaa, Whitworth & Meechan (2012a)	United Kingdom	Double-blind randomized trial	Web-based program for randomization	38/35	Over 16 years		Maxillary permanent teeth	Electronic pulp tester	–	4% Articaine with 1:100,000 epinephrine	2% lidocaine hydrochloride and epinephrine 1:80,000	BI	BI	2	2	10	Successful infiltration anaesthesia	BIs with 4% Articaine with 1:100,000 epinephrine and 2% lidocaine with 1:80,000 epinephrine produced similar levels of successful pulp anaesthesia, similar onset times of successful pulp anaesthesia, and similar levels of pain-free treatment in patients attending with irreversible pulpitis in the maxilla.	
Kanaa, Whitworth & Meechan (2012b)	United Kingdom	Double-blind randomized trial	Web-based program for randomization	25/25	18 or older	–	Mandibular teeth	Electronic pulp tester	–	4% Articaine HCL with epinephrine 1:100,000	2% Lidocaine HCL with 1:80,000 epinephrine	Supplementary BI after IANB failure	Repeat lidocaine IANB after IANB failure	2.0	2.0	5	Successful infiltration anaesthesia	BI of 4% Articaine with epinephrine allowed more pain-free treatments than repeat IANB injections for patients experiencing irreversible pulpitis in mandibular permanent teeth.	
Subbiya et al. (2011)	India	Clinical study	Non- random	30/30	37 years	–	Symptomatic mandibular first molar	Cold testing with an ice stick Electric pulp tester	No/-	4% Articaine with 1:2,00,000 adrenaline	2% Lignocaine with 1:2,00,000	BI injection	IANB	1.7	1.7	15	Aesthetic success	4% Articaine with 1:1,00,000 adrenaline can be considered as an alternative for anesthetising mandibular first molar instead of IANB with 2% lignocaine with 1:2,00,000 adrenaline.	
Aggarwal et al. (2011)	India	Prospective, randomized, double-blind study	Simple random generator	24/24	31/ 30. 4	11 M, 13 F/ 12 M, 11 F	Mandibular molar	Cold testing with an ice stick and an electric pulp tester	No/Yes	4% Articaine with 1:100,000 ephinephrine	2% Lidocaine with 1:200,000 epinephrine	IANB plus BI	IANB injections	Standard 4% Articaine cartridge	1.8	15	Pre-post-operative pain Success of anaesthesia	Supplemental infiltrations of Articaine along with conventional IANB, can be a useful adjunct in management of odontogenic pain in irreversible pulpitis	
Poorni et al. (2011)	India	Randomized double-blind clinical trial	Simple randomization	Test arm A 52 Test arm B 52 Test arm C 52	Test arm A 24.40 (4.19) Test arm B 23.46(3.7) Test arm C 24.13(4.21)	Test arm A 28 M 24 F Test arm B 30 M, 32 F Test arm C 32 M, 20 F	Symptomatic Mandibular molars	Cold testing with an ice stick Electric pulp tester	No/Yes	4% Articaine with 1:100,000 epinephrine	2% Lidocaine with 1:100,000	Test arm A IANB with 4% Articaine Test arm B BI with 4% Articaine	Test arm C IANB with 4% Lidocaine	1.8	1.8	20	Successful anaesthesia	There is no statistically significant difference among IANB and infiltration of Articaine when compared with IANB of lidocaine in mandibular molars with irreversible pulpitis.	
Notes.

BI BI

IANB Inferior Alveolar Nerve Block

F Female

M Male

SD Standard deviation

A significant methodological heterogeneity was observed between the studies. This could be attributed due to the differences in the anatomic location of the teeth being anaesthetized (mandible (Gao & Meng, 2020; Kumar et al., 2020; Aggarwal et al., 2019; Ghazalgoo et al., 2018; Martínez-Martínez, Freyle-Granados & Senior-Carmona, 2018; Shapiro et al., 2018; Aggarwal, Singla & Miglani, 2017; Umesh, 2017; Allegretti et al., 2016; Zain et al., 2016; Monteiro et al., 2015; Rajput et al., 2015; Ahmad et al., 2014; Rogers et al., 2014; Sood, Hans & Shetty, 2014; Ashraf et al., 2013; Tortamano et al., 2013; Kanaa, Whitworth & Meechan, 2012b; Subbiya et al., 2011; Aggarwal et al., 2011; Poorni et al., 2011) or maxilla (Hosseini et al., 2016; Saraf et al., 2016; Nabeel, Ahmed & Sikander, 2014; Kanaa, Whitworth & Meechan, 2012a), anterior (Saraf et al., 2016; Kanaa, Whitworth & Meechan, 2012a; Kanaa, Whitworth & Meechan, 2012b) or posterior (Gao & Meng, 2020; Kumar et al., 2020; Aggarwal et al., 2019; Ghazalgoo et al., 2018; Martínez-Martínez, Freyle-Granados & Senior-Carmona, 2018; Shapiro et al., 2018; Aggarwal, Singla & Miglani, 2017; Umesh, 2017; Allegretti et al., 2016; Hosseini et al., 2016; Saraf et al., 2016; Zain et al., 2016; Monteiro et al., 2015; Rajput et al., 2015; Ahmad et al., 2014; Nabeel, Ahmed & Sikander, 2014; Rogers et al., 2014; Sood, Hans & Shetty, 2014; Ashraf et al., 2013; Tortamano et al., 2013; Kanaa, Whitworth & Meechan, 2012a; Kanaa, Whitworth & Meechan, 2012b; Aggarwal et al., 2011; Poorni et al., 2011), tooth type (molars Gao & Meng, 2020; Kumar et al., 2020; Aggarwal et al., 2019; Ghazalgoo et al., 2018; Martínez-Martínez, Freyle-Granados & Senior-Carmona, 2018; Shapiro et al., 2018; Aggarwal, Singla & Miglani, 2017; Umesh, 2017; Allegretti et al., 2016; Hosseini et al., 2016; Zain et al., 2016; Monteiro et al., 2015; Rajput et al., 2015; Ahmad et al., 2014; Rogers et al., 2014; Sood, Hans & Shetty, 2014; Ashraf et al., 2013; Tortamano et al., 2013; Kanaa, Whitworth & Meechan, 2012b; Subbiya et al., 2011; Aggarwal et al., 2011; Poorni et al., 2011, premolars (Saraf et al., 2016; Nabeel, Ahmed & Sikander, 2014; Kanaa, Whitworth & Meechan, 2012a; Kanaa, Whitworth & Meechan, 2012b) or anterior teeth (Saraf et al., 2016; Kanaa, Whitworth & Meechan, 2012a; Kanaa, Whitworth & Meechan, 2012b)), volume of anaesthetic solution administered during the intervention (0.6 mL (Aggarwal et al., 2019), 0.9 mL (Gao & Meng, 2020), 1.7 mL (Shapiro et al., 2018; Rajput et al., 2015; Nabeel, Ahmed & Sikander, 2014; Rogers et al., 2014; Subbiya et al., 2011), 1.8 mL (Kumar et al., 2020; Martínez-Martínez, Freyle-Granados & Senior-Carmona, 2018; Aggarwal, Singla & Miglani, 2017; Hosseini et al., 2016; Zain et al., 2016; Monteiro et al., 2015; Sood, Hans & Shetty, 2014; Ashraf et al., 2013; Tortamano et al., 2013; Aggarwal et al., 2011; Poorni et al., 2011), 2.0 mL (Lokhande et al., 2019; Saraf et al., 2016; Kanaa, Whitworth & Meechan, 2012a; Kanaa, Whitworth & Meechan, 2012b), 3 mL (Umesh, 2017), 3.6 mL (Allegretti et al., 2016) and 6 mL (Umesh, 2017)), concentration of epinephrine in articaine and lidocaine and delivery route of the anaesthetic solution. Interestingly, the anaesthetic solutions for mandibular teeth were delivered via IANB (Kumar et al., 2020; Ghazalgoo et al., 2018; Martínez-Martínez, Freyle-Granados & Senior-Carmona, 2018; Aggarwal, Singla & Miglani, 2017; Umesh, 2017; Allegretti et al., 2016; Ahmad et al., 2014; Sood, Hans & Shetty, 2014; Tortamano et al., 2013; Poorni et al., 2011), BI (Hosseini et al., 2016; Zain et al., 2016; Monteiro et al., 2015; Rajput et al., 2015; Nabeel, Ahmed & Sikander, 2014; Subbiya et al., 2011; Poorni et al., 2011), supplementary BI after IANB failure (Gao & Meng, 2020; Shapiro et al., 2018; Rogers et al., 2014; Kanaa, Whitworth & Meechan, 2012b), BI combined with intraligamentary injection (Lokhande et al., 2019), supplementary intraligamentary injections after IANB failure (Aggarwal et al., 2019), standard IANB and long Bis (Ashraf et al., 2013) and IANB plus BI (Aggarwal et al., 2011) and for maxillary teeth via BI (Kanaa, Whitworth & Meechan, 2012a), anterior middle superior alveolar nerve block (AMSA) (Saraf et al., 2016) and infraorbital nerve block (IONB) (Saraf et al., 2016). In nineteen studies, both the anaesthetic solutions were administered by the same dentist. (Gao & Meng, 2020; Kumar et al., 2020; Ghazalgoo et al., 2018; Lokhande et al., 2019; Martínez-Martínez, Freyle-Granados & Senior-Carmona, 2018; Shapiro et al., 2018; Aggarwal, Singla & Miglani, 2017; Umesh, 2017; Allegretti et al., 2016; Zain et al., 2016; Ahmad et al., 2014; Rogers et al., 2014; Sood, Hans & Shetty, 2014; Ashraf et al., 2013; Kanaa, Whitworth & Meechan, 2012a; Kanaa, Whitworth & Meechan, 2012b; Aggarwal et al., 2011; Poorni et al., 2011) (Table 2).

The outcome parameters assessed post-intervention varied across studies. The success of pulpal anaesthesia assessed by VAS on a 4-point Likert scale or on a measured scale as no/mild pain during root canal treatment (endodontic access and pulpectomy) after administration of local anaesthesia, whereas the postoperative pain was assessed by the corresponding pain score on the numerical scale (Shapiro et al., 2018; Saraf et al., 2016; Zain et al., 2016; Rogers et al., 2014; Subbiya et al., 2011). The onset of anaesthesia was assessed from the time lapse between the end of NB and the onset of symptoms of subjective anaesthesia, such as the feeling of heaviness at the site of injection and lip numbness, or by electric pulp testing, cold testing, pulp evaluation and canal preparation using a standard digital stop clock. For the mandibular region, 19 studies assessed the success rate of anaesthesia (Gao & Meng, 2020; Kumar et al., 2020; Aggarwal et al., 2019; Ghazalgoo et al., 2018; Lokhande et al., 2019; Martínez-Martínez, Freyle-Granados & Senior-Carmona, 2018; Shapiro et al., 2018; Aggarwal, Singla & Miglani, 2017; Allegretti et al., 2016; Zain et al., 2016; Monteiro et al., 2015; Rajput et al., 2015; Ahmad et al., 2014; Rogers et al., 2014; Sood, Hans & Shetty, 2014; Ashraf et al., 2013; Kanaa, Whitworth & Meechan, 2012b; Subbiya et al., 2011; Aggarwal et al., 2011; Poorni et al., 2011), 4 studies evaluated the postoperative pain (Umesh, 2017; Zain et al., 2016; Rogers et al., 2014; Aggarwal et al., 2011) and 1 study evaluated the mean onset time of anaesthesia (Tortamano et al., 2013). For the maxillary region, the success rate of anaesthesia was reported by 3 studies (Hosseini et al., 2016; Nabeel, Ahmed & Sikander, 2014; Kanaa, Whitworth & Meechan, 2012a), 1 study (Saraf et al., 2016) reported postoperative pain and 2 studies reported (Saraf et al., 2016; Kanaa, Whitworth & Meechan, 2012a) the mean onset time of anaesthesia. The recording time after LA injection ranged from 5–30 min in all the included studies, except in one (Umesh, 2017), which did not mention the time. VAS was used to assess the outcome in all studies except that the verbal analogue scale was applied by Allegretti et al. (2016) and the study by Tortamano et al. (2013) calculated the onset of pulpal anaesthesia in minutes. Overall, post-intervention results showed improvements in the intervention groups based on the outcome parameters (Gao & Meng, 2020; Kumar et al., 2020; Ghazalgoo et al., 2018; Lokhande et al., 2019; Martínez-Martínez, Freyle-Granados & Senior-Carmona, 2018; Shapiro et al., 2018; Aggarwal, Singla & Miglani, 2017; Umesh, 2017; Hosseini et al., 2016; Saraf et al., 2016; Zain et al., 2016; Monteiro et al., 2015; Rajput et al., 2015; Ahmad et al., 2014; Nabeel, Ahmed & Sikander, 2014; Rogers et al., 2014; Sood, Hans & Shetty, 2014; Ashraf et al., 2013; Tortamano et al., 2013; Kanaa, Whitworth & Meechan, 2012a; Kanaa, Whitworth & Meechan, 2012b; Aggarwal et al., 2011; Poorni et al., 2011).

The mandibular postoperative pain assessed by Ghazalgoo et al. (2018), Umesh (2017), Zain et al. (2016), Rogers et al. (2014) and Aggarwal et al. (2011) using Heft Parker VAS did not show a statistically significant difference between the two groups. The time point for outcome measurement and the outcomes of interest, based on the scoring criteria, were different; also, the method of delivery of the anaesthetic solution varied in each of the included studies. Thus, these were precluded from the meta-analysis, and only qualitative analysis was conducted (Table 2).

The study by Saraf et al. (2016) concluded that 4% articaine was more efficacious than 2% lidocaine, and AMSANB was more beneficial than IONB in achieving the anaesthetic effect of maxillary anterior teeth and premolars. It was a single study included for assessing the postoperative pain using the visual analogue scale and the onset time of anaesthesia for maxillary teeth and was not considered for qualitative synthesis. The onset time of anaesthesia for mandibular teeth was assessed by Tortamano et al. (2013) and concluded that 4% articaine exhibited rapid onset with the highest duration of pulpal anaesthesia in IANB. However, since this was a single study, it was not considered for quantitative synthesis (Table 2).

Assessments of the level of evidence, risk of bias, and quality

According to JBI level of evidence, 21 studies (Gao & Meng, 2020; Kumar et al., 2020; Aggarwal et al., 2019; Ghazalgoo et al., 2018; Martínez-Martínez, Freyle-Granados & Senior-Carmona, 2018; Shapiro et al., 2018; Aggarwal, Singla & Miglani, 2017; Umesh, 2017; Allegretti et al., 2016; Hosseini et al., 2016; Zain et al., 2016; Monteiro et al., 2015; Rajput et al., 2015; Nabeel, Ahmed & Sikander, 2014; Rogers et al., 2014; Sood, Hans & Shetty, 2014; Ashraf et al., 2013; Kanaa, Whitworth & Meechan, 2012a; Kanaa, Whitworth & Meechan, 2012b; Aggarwal et al., 2011; Poorni et al., 2011) were ranked at 1c and the remaining 5 studies (Lokhande et al., 2019; Saraf et al., 2016; Ahmad et al., 2014; Tortamano et al., 2013; Subbiya et al., 2011) as 1d.

The quality assessment of 11 RCTs was executed according to the Cochrane Risk of Bias Tool (Gao & Meng, 2020; Kumar et al., 2020; Aggarwal et al., 2019; Ghazalgoo et al., 2018; Lokhande et al., 2019; Martínez-Martínez, Freyle-Granados & Senior-Carmona, 2018; Shapiro et al., 2018; Aggarwal, Singla & Miglani, 2017; Umesh, 2017; Allegretti et al., 2016; Hosseini et al., 2016; Saraf et al., 2016; Monteiro et al., 2015; Rajput et al., 2015; Nabeel, Ahmed & Sikander, 2014; Rogers et al., 2014; Sood, Hans & Shetty, 2014; Ashraf et al., 2013; Tortamano et al., 2013; Kanaa, Whitworth & Meechan, 2012a; Kanaa, Whitworth & Meechan, 2012b; Aggarwal et al., 2011; Poorni et al., 2011). Moreover, 13 studies (Gao & Meng, 2020; Aggarwal et al., 2019; Martínez-Martínez, Freyle-Granados & Senior-Carmona, 2018; Shapiro et al., 2018; Aggarwal, Singla & Miglani, 2017; Umesh, 2017; Allegretti et al., 2016; Hosseini et al., 2016; Monteiro et al., 2015; Rogers et al., 2014; Ashraf et al., 2013; Kanaa, Whitworth & Meechan, 2012a; Aggarwal et al., 2011; Poorni et al., 2011) showed a low potential risk of bias, 3 (Kumar et al., 2020; Monteiro et al., 2015; Sood, Hans & Shetty, 2014) presented a moderate risk and 7 (Lokhande et al., 2019; Saraf et al., 2016; Zain et al., 2016; Rajput et al., 2015; Nabeel, Ahmed & Sikander, 2014; Tortamano et al., 2013; Kanaa, Whitworth & Meechan, 2012b) had a high potential risk of bias (Fig. 2).

Figure 2 Risk of bias summary: (A) Review authors’ judgements about each risk of bias item for each included study, (B) review authors’ judgements about each risk of bias item presented as percentages across all included studies.

MINORS was used for quality assessment of 2 non-randomised comparative studies (Ahmad et al., 2014; Subbiya et al., 2011) that presented a score of 21 (Table 3). Only 5/26 studies (Hosseini et al., 2016; Ahmad et al., 2014; Rogers et al., 2014; Ashraf et al., 2013; Poorni et al., 2011) disclosed receiving financial support for the work.

Table 3 Methodological index for non-randomized studies (MINORS).

	A clearly stated aim	Inclusion of consecutive patients	Prospective collection of data	Endpoints appropriate to the aim of the study	Unbiased assessment of the study endpoint	Follow-up period appropriate to the aim of the study	Loss to follow up less than 5%	Prospective calculation of the study size	bAn adequate control group	bContemporary groups	bBaseline equivalence of groups	*Adequate statistical analyses	Total	
Ahmad et al. (2014)	2	2	2	2	1	2	2	0	2	2	2	2	21	
Subbiya et al. (2011)	2	2	2	2	1	2	2	0	2	2	2	2	21	
Notes.

a The items are scored 0 (not reported), 1 (reported but inadequate) or 2 (reported and adequate). The global ideal score being 16 for non-comparative studies and 24 for comparative studies.

b For study with control group.

Synthesis of results

A total of 22 studies (Gao & Meng, 2020; Kumar et al., 2020; Aggarwal et al., 2019; Lokhande et al., 2019; Martínez-Martínez, Freyle-Granados & Senior-Carmona, 2018; Shapiro et al., 2018; Aggarwal, Singla & Miglani, 2017; Allegretti et al., 2016; Hosseini et al., 2016; Zain et al., 2016; Monteiro et al., 2015; Rajput et al., 2015; Ahmad et al., 2014; Nabeel, Ahmed & Sikander, 2014; Rogers et al., 2014; Sood, Hans & Shetty, 2014; Ashraf et al., 2013; Tortamano et al., 2013; Kanaa, Whitworth & Meechan, 2012a; Kanaa, Whitworth & Meechan, 2012b; Subbiya et al., 2011; Aggarwal et al., 2011; Poorni et al., 2011) fulfilled the inclusion criteria for quantitative analysis. Subsequently, two meta-analyses, including one subgroup analysis, were performed on the success rate of anaesthesia for mandibular and maxillary teeth.

The success rate of anaesthesia for mandibular teeth

The pooled outcomes from 19 studies, in tooth level unit, using random-effect model showed that 4% articaine resulted in a success rate 1.37-fold higher than that of 2% lidocaine (RR, 1.37; 95% CI [1.17–1.62]; P = 0.0002), showing statistically significant difference favouring the articaine group and 72% heterogeneity (I2). When subgroup analysis was performed according to the injection techniques using random-effect model, it was observed that for IANB (Kumar et al., 2020; Martínez-Martínez, Freyle-Granados & Senior-Carmona, 2018; Aggarwal, Singla & Miglani, 2017; Allegretti et al., 2016; Ahmad et al., 2014; Sood, Hans & Shetty, 2014; Poorni et al., 2011), 4% articaine resulted in a success rate 1.25-fold higher than that for 2% lidocaine (RR, 1.25; 95% CI [0.98–1.59]; P = 0.007) showing statistically significant difference favouring the articaine group with 43% heterogeneity. For BI technique (Kumar et al., 2020; Zain et al., 2016; Monteiro et al., 2015; Rajput et al., 2015; Kanaa, Whitworth & Meechan, 2012b; Poorni et al., 2011), 4% articaine resulted in a success rate 1.13-fold higher than that for 2% lidocaine (RR, 1.13; 95% CI [0.89–1.45]; P = 0.31) with 68% heterogeneity. Supplementary BI after IANB failure technique (Gao & Meng, 2020; Shapiro et al., 2018; Rogers et al., 2014; Kanaa, Whitworth & Meechan, 2012b) showed that 4% articaine resulted in a success rate of 1.53-fold higher than 2% lidocaine (RR, 1.53; 95% CI [1.821–1.94]; P = 0.0004) showing statistically significant difference favouring the articaine group with the heterogeneity of 36% (I2). BI combined with intraligamentary injection technique (Lokhande et al., 2019) showed that 4% articaine resulted in a success rate of 8.33-fold higher than that for 2% lidocaine (RR, 8.33; 95% CI [2.81–24.67]; P = 0.0001) showing statistically significant difference favouring the articaine group. For supplementary intraligamentary injections after IANB failure (Aggarwal et al., 2019) and standard IANB and long BI technique (Ashraf et al., 2013), 4% articaine resulted in a success rate 0.9-fold (RR, 0.9; 95% CI [0.67–1.2]; P = 0.47) and 2.41-fold (RR, 2.41; 95% CI [1.56–1.2]; P < 0.0001) higher than that of 2% lidocaine, respectively showing statistically significant difference favouring the articaine group for standard IANB and long BI technique but not for supplementary intraligamentary injections after IANB failure. For IANB plus BI technique (Aggarwal et al., 2011), 4% articaine resulted in a success rate of 1.38-fold higher than that for 2% lidocaine (RR, 1.38; 95% CI [0.74–2.6]; P < 0.0001) showing a statistically significant difference favouring the articaine group (Fig. 3).

Figure 3 Forest plot of the pooled analysis and the subgroup analysis comparing 4% articaine with 2% lidocaine in the clinical success of mandibular block and infiltration anaesthesia for irreversible pulpitis in tooth unit.

The success rate of anaesthesia for maxillary teeth

The pooled outcomes from 3 studies (Hosseini et al., 2016; Nabeel, Ahmed & Sikander, 2014; Kanaa, Whitworth & Meechan, 2012a) on tooth level unit for maxillary buccal infiltration technique showed that 4% articaine resulted in a success rate of 1.06-fold higher than that for 2% lidocaine (RR, 1.06; 95% CI [0.95–1.2]; P = 0.3, I2 = 0%) using random-effects model, showing statistically insignificant difference (Fig. 4).

Figure 4 Forest plot comparing 4% with 2% lidocaine in the clinical success rate of maxillary buccal infiltration for irreversible pulpitis in tooth unit.

Sensitivity analysis

Table 4 represents the results of sensitivity analysis for mandibular teeth. Studies of high risk (Lokhande et al., 2019; Saraf et al., 2016; Zain et al., 2016; Rajput et al., 2015; Nabeel, Ahmed & Sikander, 2014; Tortamano et al., 2013; Kanaa, Whitworth & Meechan, 2012b), moderate risk (Kumar et al., 2020; Monteiro et al., 2015; Sood, Hans & Shetty, 2014) or low risk of bias (Gao & Meng, 2020; Kumar et al., 2020; Aggarwal et al., 2019; Ghazalgoo et al., 2018; Lokhande et al., 2019; Martínez-Martínez, Freyle-Granados & Senior-Carmona, 2018; Shapiro et al., 2018; Aggarwal, Singla & Miglani, 2017; Umesh, 2017; Allegretti et al., 2016; Hosseini et al., 2016; Saraf et al., 2016; Monteiro et al., 2015; Rajput et al., 2015; Ahmad et al., 2014; Nabeel, Ahmed & Sikander, 2014; Rogers et al., 2014; Sood, Hans & Shetty, 2014; Ashraf et al., 2013; Tortamano et al., 2013; Kanaa, Whitworth & Meechan, 2012a; Kanaa, Whitworth & Meechan, 2012b; Aggarwal et al., 2011; Poorni et al., 2011) were excluded from sensitivity analysis. The clinical success rates comparing the articaine and lidocaine groups showed a significant change after the exclusion of these studies. Reanalysis using the fixed-effect model also showed that the outcomes were not adverse. The exclusion of subgroups with a single study showed a significant improvement in the success rate of articaine group as compared to the lidocaine group in the treatment of IP. Moreover, the subgroup analysis for different injection techniques showed a significant change after the inclusion of only supplementary BI after IANB failure technique as compared to the inclusion of IANB and BI technique (Table 4).

Table 4 Sensitivity and subgroup analysis of the outcomes between articaine and lidocaine group in tooth unit for mandibular region.

Item	Success of local anaesthesia (RR, 95% CI)	
Original estimates	1.37 [1.17, 1.62], P = 0.0002	
Exclusion all the studies of high risk of bias	1.28 [1.09, 1.52], P = 0.003	
Exclusion all the studies of moderate risk of bias	1.41 [1.17, 1.70], P = 0.0003	
Exclusion all the studies of low risk of bias	1.58 [1.12, 2.24], P = 0.003	
Inclusion of studies of low risk of bias only	1.31 [1.08, 1.60], P = 0.003	
Fixed or random effects	
Fixed effect	1.37 [1.26, 1.49], P < 0.00001	
Random effect	1.37 [1.17, 1.62], P = 0.0002	
Exclusion of subgroups with single study	1.30 [1.19, 1.41], P = 0.001	
Inclusion of IANB only	1.25 [0.98, 1.59], P = 0.07	
Inclusion of buccal infiltration Only	1.13 [0.89, 1.45], P = 0.31	
Inclusion of supplementary buccal infiltration after IANB failure Only	1.53 [1.21, 1.94], P = 0.0004	
Notes.

CI Confidence interval

IANB Inferior alveolar nerve block

RR Relative risk

Publication bias for studies included on mandibular anaesthetic technique was evaluated using a funnel plot (Fig. 5). The funnel plot showed asymmetry at the apex from the centre line having more studies on the right side as compared to left, representing a lack of inclusion of publications describing non-significant intervention results as well as the omission of unpublished studies that might result in an over-estimation of the true effect of an intervention.

Figure 5 Funnel plot comparing 4% articaine with 2% lidocaine in the clinical success of mandibular block and infiltration anaesthesia for irreversible pulpitis in tooth unit.

Discussion

Endodontic pain management is a critical component in reducing extreme anxiety during endodontic treatment. LA’s in-depth and methodical expertise, as well as its suitable delivery methodologies, are essential for pain-free dental treatment (Aggarwal et al., 2011). The gold standard amide anaesthetic, lidocaine, has a brief start of action, but when combined with epinephrine, the duration of action increases to intermediate (Ghazalgoo et al., 2018). In patients with endodontic discomfort, the success of lidocaine in IANB and infiltration remains low (Aggarwal et al., 2011; Aggarwal, Jain & Debipada, 2009; Hargreaves & Keiser, 2002). The risk of failed local anaesthetic was eight times greater in patients with IP than in normal individuals (Hargreaves & Keiser, 2002). Pain is initially transmitted by A-delta and C-fibres in IP, but as the inflammatory process advances, C-fibre transmission takes over, resulting in changed pain characteristics. Strong, quick, acute, and well-localized pain is caused by A-delta fibres, whereas dull, persistent, and radiating pain is caused by C-fibres (Prpic-Mehicic & Nada, 2010).

Anatomical causes, acute tachyphylaxis, and the influence of inflammation on local tissue pH, blood flow, nociceptors, central sensitization, and psychological variables are all possible reasons for failure (Aggarwal et al., 2011; Poorni et al., 2011; Aggarwal, Jain & Debipada, 2009; Reader, Nusstein & Hargreaves, 2011). Articaine is more effective in reducing the action potential produced by A-fibres as compared to 2% lidocaine and the complete disappearance of the action potential produced by C-fibres (Allegretti et al., 2016) (Fig. 6).

Figure 6 The chemical structure of lidocaine and articaine.

This systematic review and meta-analysis of clinical trials provides level 1 evidence for evaluating the efficacy of 4% articaine and 2% lidocaine in the mandibular and maxillary block and infiltration anaesthesia in patients with IP, according to the Oxford Centre for Evidence-based Medicine’s levels of evidence criteria (Oxford Centre for Evidence-Based Medicine, 2009). The review included 26 research published between 2011 and 2020 in various countries (24 randomised trials and 2 non-randomised trials). The study participants ranged in age from 15 to 65 years old and were of both genders. Therefore, the findings of this systematic review may be applied to a wide variety of people, as well as anaesthetic effectiveness in mandibular and maxillary teeth with symptomatic IP. Pulp sensitivity tests and the exclusion of periapical diseases were used to determine if research subjects met the criteria for symptomatic IP in the included studies. Participants who have been taking any medication that may affect the impact of local anaesthetic were also excluded from the research, reducing selection bias.

The most common method for anaesthetizing mandibular teeth is IANB (Zain et al., 2016). However, IANB is an unreliable anaesthetic method, especially in the case of IP; even when correctly administered, the success rate varied from 15–25% (Kumar et al., 2020; Shapiro et al., 2018; Zain et al., 2016). This emphasises the need of having alternatives to IANB (Shapiro et al., 2018). Thus, in the present systematic review, the pulpal anaesthetic efficacy between articaine and lidocaine were assessed using various infiltration and block techniques. For mandibular teeth IANB, BI, supplementary BI, and intraligamentary injections after IANB failure, BI combined with intraligamentary injection, standard IANB, and long BIs, IANB plus BI and for maxillary teeth via BI, AMSA and IONB were used. The 2011 meta-analysis by Brandt et al. (2011) found that articaine as an infiltrating agent was 3.8-fold more likely to be successful than lidocaine. A recent meta-analysis by Kung, McDonagh & Sedgley (2015) demonstrated that in cases of a failed IANB, supplementary infiltration with 4% articaine was 3.55-fold more successful in achieving profound anaesthetic effect than 2% lidocaine.

Various criteria were applied in this review to assess the first pulpal anaesthetic success, including lip numbness, cold testing, and electric pulp testing. Lip numbness was noted in the majority of the trials, albeit it was a subjective symptom. Only a few studies used a cold test or an electric pulp test followed by lip numbness to validate the first pulpal anaesthetic success, as there was a weak connection between lip anaesthesia, cold testing, and pulpal anaesthesia following IANB for IP mandibular molars (Shapiro et al., 2018; Umesh, 2017). Bjorn was the first to link a negative response to the maximal output of electrical pulp stimulation for painless dental treatment (Bjorn, 1946). Dreven et al. (1987) examined the reaction to an electric pulp tester as a measure of pulpal anaesthetic prior to endodontic treatment in teeth with normal pulp, reversible pulpitis, and irreversible pulpitis. In IP, however, a lack of reaction to cold or electric pulp tests does not always imply pulpal anaesthesia (Dreven et al., 1987). This might be because in teeth with IP, the reactions to electric pulp tests and cold testing are linked to rapid and slow silent A-delta fibres, respectively.

Therefore, it can be assumed that if the tetrodotoxin-resistant sodium channels appear on deeper nociceptive C fibres, then neither negative nor positive responses to EPT and cold tests indicate the success of anaesthesia as the C fibres might be accountable for the pain response (Shapiro et al., 2018; Brandt et al., 2011). Hence, an appropriate alternative is to record the pain response during access cavity opening and pulp extirpation. All the studies included in the systematic review assessed the clinical success of 4% articaine and 2% lidocaine based on pain response during access preparation using VAS.

In this systematic review, the efficacy of anaesthetic solutions was measured by clinical success rate, postoperative discomfort, and local anaesthesia onset time. The meta-analysis of the collected data from 22 studies that satisfied the inclusion criteria was based on the clinical success rate. Furthermore, there was methodological variability in terms of research location, study setting, sample size, number and expertise of investigators performing procedures and diagnosis, volume of LA solution, epinephrine concentration, and marking on the pain scale. This heterogeneity was addressed by using a random-effects model for meta-analysis (Kung, McDonagh & Sedgley, 2015). This meta-analysis found that when used in mandibular and maxillary block and infiltration anaesthesia, 4% articaine, which is a more concentrated LA solution, was more likely to provide anaesthetic success than less concentrated 2% lidocaine anaesthetic solution. Besides, wide CIs were observed in the forest plot analysis, potentially contributing to heterogeneity as shown by I2 estimates, i.e., 72%, with statistically significant difference favouring the articaine group for mandibular teeth (Fig. 3) as compared to narrower CIs contributing to 0% heterogeneity favouring the articiane group with non-significant difference favouring the articaine group for maxillary teeth (Fig. 4). Similar results were reported by previous meta-analysis that were conducted (Su et al., 2016; De Geus et al., 2020; Kung, McDonagh & Sedgley, 2015; Brandt et al., 2011). Discrepancies in the chemical properties of the molecular structures of the 4 percent articaine and 2 percent lidocaine LA agents might have resulted in clinical differences. The uncharged form of a LA molecule is required for diffusion across the sheaths of lipid neurons and cell membranes, therefore the anaesthetic dissociation constant (pKa) is a crucial number for successful anaesthesia (De Geus et al., 2020). In addition to ionisation, fat solubility and protein-binding properties contribute to the clinical characteristics of LAs whereas, their clinical performance is influenced by the site of injection, concentration of drug and vasoconstrictor, injection volume, and anaesthetic solution’s inherent vasodilatory properties (De Geus et al., 2020; Haas, 2002; Moore & Hersh, 2010). A previous study showed that articaine suppresses the compound action potential of the A fibres in the isolated rat sural nerve (Potonik et al., 2006). Also, ionic channels are blocked even in lower concentrations with the thiophene derivative (articaine) as compared to the benzene derivative (lidocaine) (Kolli, Nirmala & Nuvvula, 2017).

The differences in the method of delivery of LA for mandibular teeth for the included trials were one of the major challenges posed by this meta-analysis. In the sub-group analysis, a significant advantage of using articaine over lidocaine for supplementary infiltration after mandibular block anaesthesia over the mandibular block and infiltration anaesthesia alone was observed with heterogeneity of 36% (Fig. 3). Another meta-analysis by Brandt et al. (2011) showed that the pulpal anaesthetic efficacy of articaine was markedly superior to lidocaine when used during infiltration. One study each for supplementary intraligamentary injection, IANB plus long buccal, and IANB with BI was included in the subgroup analysis for mandibular teeth. However, to evaluate whether the final results were dependent on subgroup results of these single studies, a sensitivity analysis was performed via meta-analysis by excluding the above studies in question. This analysis confirmed that although the exclusion of the studies reduced the RRs and heterogeneity, the overall results were unchanged (Table 4).

The secondary objective of this systematic review was to assess the postoperative pain and mean onset time of the two LA solutions. The group receiving 4% articaine, as opposed to 2% lidocaine, experienced less pain as measured by VAS during the injection and treatment phases, which might be due to articaine’s 1.5-fold higher potency than lidocaine’s (Su et al., 2016). In terms of onset time, 4 percent articaine was shown to be faster than 2 percent lidocaine in pulpal anaesthesia. This phenomenon might be explained by the fact that the onset period of anaesthesia is proportional to the pace of epineural diffusion. This rate is proportional to the percentage of drug in the base form, which is proportional to the pKa; articaine’s pKa was lower than lidocaine’s (Su et al., 2016).

Intriguingly, when this study was compared to prior English language systematic reviews (Su et al., 2016; Kung, McDonagh & Sedgley, 2015; Brandt et al., 2011; Katyal, 2010), there were some striking similarities and variations in terms of anaesthetic solution and administration techniques. Except for the reviews by Su et al. (2016) and Kung et al. (Brandt et al. 2011), the main difference between this and previous reviews (Su et al., 2016; Kung, McDonagh & Sedgley, 2015; Brandt et al., 2011; Katyal, 2010) is that all subjects in this review were diagnosed with IP, whereas previous reviews consisted of a broad cohort of patients and non-patient volunteers with or without pain (Srinivasan et al., 2017; Kung, McDonagh & Sedgley, 2015). In comparison to the current analysis, which comprised parallel-design clinical trials, earlier reviews by Brandt et al. (2011) and Katyal (2010) used crossover design. In comparison to prior research, the start and end of the search time changed in the current study. Previous studies mostly looked at adverse events, pain, and the onset of local anaesthetic, but this study examined at the total clinical success rate as well as subgroup and sensitivity analyses.

Nevertheless, the present review has some limitations. It was not possible to fully avoid the clinical heterogeneity among the included studies. The sample size of the studies was small, thus lacking statistical power. The reasons for the difference in postoperative pain and the onset time could not be explained because of lack of evidence. Individual tooth type analysis (incisors, canines, premolars, and molars) was not performed, and age and gender were also not taken into consideration in the analysis. However, 15 studies rated good on methodological validity assessment, exhibiting a low risk of bias. The subgroup and sensitivity analyses were performed to rule out the potential reasons for heterogeneity. Thus, it is suggested that in the future, high-quality clinical trials on the outcomes of onset of anaesthesia and pain assessment at various stages of the treatment procedure of IP should be conducted along with the trials assessing the adverse effects of the two solutions at varying concentrations and sites of injection.

Conclusions

The current systematic review and meta-analysis found that whether administered for mandibular block or maxillary infiltration in patients with symptomatic IP, articaine is superior to lidocaine. For mandibular teeth, 4 percent articaine had a clinical success rate 1.37 times greater than 2 percent lidocaine, and 1.06 times higher for maxillary teeth.

KEY HIGHLIGHTS of the current analysis

1. Successful pulpal anaesthesia is the cornerstone for painless root canal treatment, especially in patients with symptomatic pulpitis.

2. Articaine was introduced to overcome supplemental anaesthesia and to increase the effectiveness of the quality of anaesthesia.

3. Articaine is associated with a lower visual analogue scale rating for pain.

4. Articaine resulted in 1.37-fold and 1.06-fold higher clinical success rate than lidocaine for mandibular and maxillary teeth respectively.

Supplemental Information

Supplemental Information 1 PRISMA checklist

Click here for additional data file.

Supplemental Information 2 Systematic Review and Meta-Analysis Rationale and contribution of the article

Click here for additional data file.

Additional Information and Declarations

Competing Interests

Author Contributions

Data Availability

The authors declare there are no competing interests.

Sanjay Miglani conceived and designed the experiments, performed the experiments, analyzed the data, prepared figures and/or tables, and approved the final draft.

Irfan Ansari conceived and designed the experiments, analyzed the data, prepared figures and/or tables, and approved the final draft.

Swadheena Patro and Ankita Mohanty performed the experiments, prepared figures and/or tables, and approved the final draft.

Shahnaz Mansoori and Bhoomika Ahuja performed the experiments, analyzed the data, prepared figures and/or tables, and approved the final draft.

Mohmed Isaqali Karobari conceived and designed the experiments, analyzed the data, authored or reviewed drafts of the paper, and approved the final draft.

Krishna Prasad Shetty, Musab Hamed Saeed and Alexander Maniangat Luke analyzed the data, authored or reviewed drafts of the paper, and approved the final draft.

Ajinkya M. Pawar conceived and designed the experiments, performed the experiments, authored or reviewed drafts of the paper, and approved the final draft.

The following information was supplied regarding data availability:

This is a review article. There is no raw data.

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
