# Peer review of "Efficacy of 4% articaine vs 2% lidocaine in mandibular and maxillary block and infiltration anaesthesia in patients with irreversible pulpitis: a systematic review and meta-analysis"

_PeerJ, doi:10.7717/peerj.12214_

## Round 0.1 · original submission · Minor Revisions

Dear authors,

Please follow the reviewers' instructions.

Best regards

·

Basic reporting

No comment

Experimental design

No comment

Validity of the findings

No comment

Additional comments

Dear Author,
A comprehensive qualitative and quantitative analysis. The manuscript is well-drafted.
Only minor revisions/queries need to be addressed:
1) Was the review registered in any databases like PROSPERO. The importance of such registration is that it allows you to declare to the reviewers and the editor that the review protocol has been screened by PROSPERO and either there are no similar reviews or a justification is provided for the current review. Thus, I request the authors to register in PROSPERO and provide the registeration number
2) PubMed/Medline- Here PubMed included both Medline and PubMed central right? Then why mention as slash Medline?
3) Scopus database?
4) Have u included a supplementary file summarizing the search strategy employed in each database and have enlisted the articles excluded in full text assessment and the reason for exclusion?
5) Keywords and other free terms- Here what is the difference between keywords and other free terms?

Kindly address the above comments.

Reviewer 2 ·

Basic reporting

English Language should be revised.
References are well presented.

Experimental design

The experimental design of the manuscript Is ok, aim are clear and method are well described.

Validity of the findings

See General Comments

Additional comments

Dear Authors,
Thank you for submitting your article to this prestigious journal.
The manuscript is well presented but it presents some issues:

Abstract: Abstract should be revised, rewritten and make it more fluent and appealing.

Keywords: To ensure a properly research in medical databases, use MeSH terms to find appealing keywords that could be helpful in finding your article.

From Line 93 to 98… I would like to propose the authors to insert the chemical formula of lidocaine and articaine as a figure to ensure a proper comprehension for the readers.
Moreover could you please describe the pharmacodynamics?
Figures are well presented.
Table 2: if it is possible make it more clear. It is too confusing.
Methods and Results are ok.
From Line 346 to 349… Please be more specific in describing the neurological patterns. Where the pain impulse is distributed in the central nervous system? What are the areas of the cortex included in the pain storage?
Discussion should be revised.
Conclusion must be entirely revised making it more appealing.
English spell revision is necessary.

Best Regards.

Annotated reviews are not available for download in order to protect the identity of reviewers who chose to remain anonymous.

---

## Round 0.2 · accepted · Accept

Dear authors thank you for submitting your manuscript to this prestigious journal.

The article according to the reviewers' assessment is now ready to be accepted.

Best regards

·

Basic reporting

No comment

Experimental design

No comment

Validity of the findings

No comment

Additional comments

The revisions are to my satisfaction. The manuscript can be accepted for publication

Reviewer 2 ·

Basic reporting

No comment

Experimental design

No comment

Validity of the findings

No comment

Additional comments

Dear authors thank you for submitting the revised version of your manuscript.
I will suggest the editor to accept it in current form
Best regards